# On the Feasibility of Simple Transformer for Dynamic Graph Modeling

Submission Id: 1847

## ABSTRACT

Dynamic graph modeling is crucial for understanding complex structures in web graphs, spanning applications in social networks, recommender systems, and more. Most existing methods primarily emphasize structural dependencies and their temporal changes. However, these approaches often overlook detailed temporal aspects or struggle with long-term dependencies. Furthermore, many solutions overly complicate the process by emphasizing intricate module designs to capture dynamic evolutions. In this work, we harness the strength of the Transformer's self-attention mechanism, known for adeptly handling long-range dependencies in sequence modeling. Our approach offers a simple Transformer model tailored for dynamic graph modeling without complex modifications. We re-conceptualize dynamic graphs as a sequence modeling challenge and introduce an innovative temporal alignment technique. This technique not only captures the inherent temporal evolution patterns within dynamic graphs but also streamlines the modeling process of their evolution. As a result, our method becomes versatile, catering to an array of applications. Our model's effectiveness is underscored through rigorous experiments on four real-world datasets from various sectors, solidifying its potential in dynamic graph modeling. The datasets and codes are available[1].

## CCS CONCEPTS

• **Computing methodologies → Learning latent representations**; • **Information systems → Data mining**; **World Wide Web**.

## KEYWORDS

Dynamic graphs, Transformer, graph representation learning

**ACM Reference Format:**

Anonymous Author(s). 2024. On the Feasibility of Simple Transformer for Dynamic Graph Modeling. In *Proceedings of Make sure to enter the correct conference title from your rights confirmation emai (Conference acronym 'XX).* ACM, New York, NY, USA, 10 pages. https://doi.org/XXXXXXX.XXXXXXX

## 1 INTRODUCTION

Graph-structured data are prevalent on the World Wide Web, such as social networks [9, 32], recommender systems [38, 42], article

---

[1]https://anonymous.4open.science/r/SimpleDyG/

citation graphs [15, 47], dialogue systems [21, 23], and so on. Thus, graph-based mining and learning have become fundamental tools in many Web applications, ranging from analyzing users' behaviors ranging from the message-exchanging within social friendships, ratings/reviews on recommender platforms, publication/citation trends in the academic community, to multi-turn task-oriented dialogue. Traditionally, many works focus on static graphs characterized by fixed nodes and edges. However, many real-world graphs on the Web are intrinsically dynamic in nature, which continuously evolve over time [36]. That is, the nodes and their edges in such graphs are undergoing constant addition or reorganization based on some underlying patterns of evolution. For example, in a social network like UCI [30], where nodes represent users and edges represent friend connections, users frequently exchange messages with their friends, and the social graph structure is constantly changing as new friendships are formed. To study this important class of graphs and their applications on the Web, we focus on dynamic graph modeling in this paper, aiming to capture the evolving patterns in a dynamic graph.

Existing works for dynamic graph modeling mainly fall into two categories: discrete-time approaches [31, 36] and continuous-time approaches [6, 40, 45, 48] as shown in Figure 1(a) and 1(b), respectively. The former regards dynamic graphs as a sequence of snapshots over a discrete set of time steps. This kind of approach usually leverages structural modules such as graph neural networks (GNN) [46] to capture the topological information of graphs, and temporal modules such as recurrent neural networks (RNN) [37] to learn the sequential evolution of dynamic graphs [36]. Meanwhile, the latter focuses on modeling continuous temporal patterns via specific temporal modules such as temporal random walk [29] or temporal kernel [7], illustrated by Figure 1(b). Despite the achievements of previous works in dynamic graphs, there still exist some key limitations. First, the modeling of temporal dynamics on graphs is still coarse-grained or short-termed. On one hand, discrete-time approaches discard the fine-grained temporal information within the snapshot, which inevitably results in partial loss of temporal patterns. On the other hand, while continuous-time approaches retain full temporal details by mapping each interaction to a continuous temporal space, capturing long-term dependency within historical graph data still remains a difficult problem [35, 50]. Second, the majority of the existing works rely extensively on the message-passing GNNs to encode the structural patterns in dynamic graphs. Although powerful in graph modeling, the message-passing mechanism shows inherent limitations such as over-smoothing [5] and over-squashing [1] that become more pronounced as model depth increases, preventing deeper and more expressive architectures.

In pursuit of addressing these limitations, we have witnessed the successful application of Transformer [41] and its variants in natural language processing (NLP) [3, 16] and computer vision (CV)

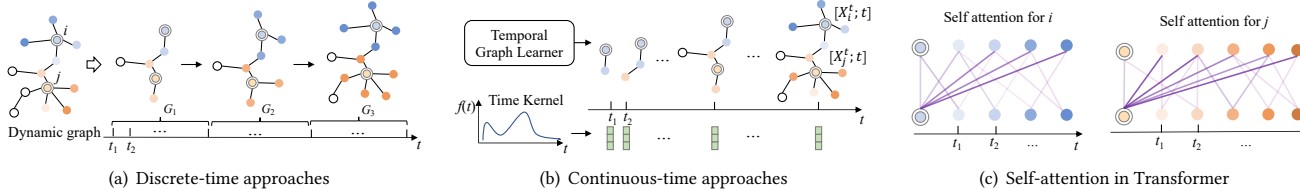

(a) Discrete-time approaches      (b) Continuous-time approaches      (c) Self-attention in Transformer

**Figure 1: Dynamic graph modeling in various ways. (a) The discrete-time approaches represent the dynamic graph into a sequence of snapshots without considering the temporal information within each snapshot. (b) The continuous-time approaches map time information of each interaction by time encoder such as time kernel. The dynamic representation of each node $X_i^t$ at time $t$ is harvest by the integration of the temporal graph learner (such as GNNs) and time feature. (c) The Transformer captures the continuous sequence of each node and the self-attention mechanism alleviates the long-term dependency issues.**

[8, 24]. The success is underpinned by two distinct advantages inherent to the Transformer architecture: as shown in Figure 1(c), it can naturally support a continuous sequence of data without the need for discrete snapshots, and its self-attention mechanism can capture long-term dependency [41], which are important factors for dynamic graph modeling. Transformers also presents a potentially better alternative to capturing topological information, as it is less or not affected by the over-smoothing and over-squashing issues associated with message-passing GNNs. Hence, in this work, we explore the feasibility of the Transformer architecture for dynamic graph modeling. In fact, we have observed a growing body of research trying to modify the Transformer for static graphs [17, 33, 49]. Nonetheless, these studies primarily focus on integrating graph structural knowledge into the vanilla Transformer model, which generally still leverage message-passing GNNs as auxiliary modules to refine positional encoding and attention matrices based on graph-derived information [27]. More recently, Ying et al. [49] indicated that the pure Transformer architecture holds promise for graphs. However, all these previous Transformer-based approaches only focus on static graphs, leaving unanswered questions about the feasibility for dynamic graphs, as we elaborate below.

The first challenge lies in the need to preserve the historical evolution throughout the entire timeline. However, due to the calculation of pairwise attention scores, existing Transformer-based graph models can only deal with a small receptive field, and would face serious scalability issues on even a moderately large graph. Notably, their primary application is limited to small-size graphs such as molecular graphs [33]. However, many dynamic graphs on the Web such as social networks or citation graphs are generally much larger for the vanilla Transformer to handle. To this end, we adopt a novel strategy wherein we treat the history of each node as a *temporal ego-graph*, serving as the receptive field of the ego-node. The *temporal ego-graph* is much smaller than the entire graph, yet it retains the full interaction history of the ego-node in the dynamic graph. Thus, we are able to preserve the temporal dynamics of every user across the entire timeline, while simultaneously ensuring scalability. Subsequently, this *temporal ego-graph* can be tokenized into a sequential input tailored for the Transformer. Remarkably, through this simple tokenization process, no modification to the original Transformer architecture is required.

The second challenge lies in the need to align temporal information across input sequences. Specifically, on dynamic graphs

different input sequences converge on a common time domain—whether absolute points in time (e.g., 10am on 12 October 2023) or relative time intervals (e.g., a one-hour time window) convey the same across all sequences generated from different nodes' history. In contrast, sequences for language modeling or static graphs lack such a universal time domain, and can be regarded as largely independent of each other. Thus, vanilla sequences without temporal alignment lack a way to differentiate variable time intervals and frequency information. For example, a bursty stream of interactions, happening over a short one-hour interval, has a distinct evolution pattern from a steady stream containing the same number of interactions, but happening over a period of one day. Therefore, it becomes imperative to introduce a mechanism that infuses temporal alignment among distinct input sequences generated from the *ego-graphs*. To address this challenge, we carefully design special *temporal tokens* to align different input sequences in the time domain. The temporal tokens serve as indicators of distinct time steps that are globally recognized across all nodes, and integrate them into the input sequences. While achieving the global alignment, local sequences from each node still retains the chronological order of the interactions in-between the temporal tokens, unlike traditional discrete-time approaches where temporal information within each snapshot is lost.

Based on the above insights, we propose a **Simple** Transformer architecture for **Dy**namic **G**raph modeling, named **SimpleDyG**. The word "simple" is a reference to the use of the original Transformer architecture without any modification, where the capability of dynamic graph modeling is simply and solely derived from constructing and modifying the input sequences. In summary, the contribution of our work is threefold.

- We explore the potential of the Transformer architecture for modeling dynamic graphs. We propose a simple yet surprisingly effective Transformer-based approach for dynamic graphs, called SimpleDyG, without complex modifications.
- We introduce a novel strategy to map a dynamic graph into a set of sequences, by considering the history of each node as a *temporal ego-graph*. Furthermore, we design special temporal tokens to achieve global temporal alignment across nodes, yet preserving the chronological order of interactions at a local level.
- We conduct extensive experiments and analysis across four real-world Web graphs, spanning diverse applications domains on the Web. The empirical results demonstrate not only the feasibility, but also the superiority of SimpleDyG.

## 2 RELATED WORK

### 2.1 Dynamic Graph Learning

Current dynamic graph learning methods can be categorized into two primary paradigms: discrete-time approaches and continuous-time approaches. In discrete-time methods, dynamic graphs are treated as a series of static graph snapshots taken at regular time intervals. To model both structural and temporal aspects, these approaches integrate the GNNs with sequence models (RNNs or self-attention mechanisms) [10, 31, 36, 39]. For instance, DySAT Sankar et al. [36] leverages Graph Attention Network (GAT) and self-attention as fundamental components for both structural and temporal modules. In contrast, EvolveGCN [31] employed an RNN to evolve the Graph Convolutional Network (GCN) parameters. Nevertheless, they often fall short of capturing the granular temporal information. Consequently, the continuous-time approaches treat the dynamic graphs as sequences of interaction events at a specific timestamp. Some approaches model dynamic graph evolution as temporal random walks or causal anonymous walks [29, 44]. Another avenue of research focuses on time window encoding techniques integrated with graph structure modeling such as temporal graph attention used in TGAT [48] and TGN [35] or MLP-Mixer layers applied in GraphMixer [6]. Additionally, researchers also leverage temporal point processes treating the arrival of nodes/edges as discrete events [14, 40, 45]. Despite the promise demonstrated by continuous-time approaches, it's important to note that they come with limitations in capturing long-term dependencies originating from historical data.

The differences between our work and the previous dynamic graph learning methods lie in two points. First, our method effectively mitigates long-term dependency challenges, leveraging the inherent advantages of the Transformer architecture. Second, our method preserves the chronological history of each ego node within the input sequences. The temporal alignment mechanisms among various ego networks empower our model to capture both global and local information within the dynamic graphs.

### 2.2 Transformers for Graphs

Transformer architectures for graphs have emerged as a compelling alternative to conventional GNNs, aiming to mitigate issues like over-smoothing and over-squashing. Prior research focused on integrating graph information into the vanilla Transformer through diverse strategies. Some methods integrate GNNs as auxiliary components to bolster structural comprehension within the Transformer architecture [18, 34]. Others focus on enriching positional embeddings by spatial information derived from the graph. For instance, Graphormer [49] integrates the centrality, spatial and edge encoding into Transformer. Cai and Lam [4] adopted distance embedding for tree-structured abstract meaning representation graph. Kreuzer et al. [19] utilized the full Laplacian spectrum to learn the positional encoding for graph. There are also studies focus on refining attention mechanisms in Transformer for graph analysis. For instance, Min et al. [28] employed a graph masking attention mechanism to seamlessly inject graph-related priors into the Transformer architecture. Excepted for the complicated design, more recently, Kim et al. [17] shed light on the effectiveness of pure Transformers in

graph learning. Their approach treats all nodes and edges as independent tokens, severing as inputs for Transformer. Recently, Mao et al. [25] proposed a Transformer based model for heterogeneous information networks. Node-level structure and heterogeneous relation are integrated into the attention mechanism.

It's worth noting that most of the previous works based on Transformers mainly focused on static graphs. Recently, Yu et al. [50] introduced a Transformer based model designed for dynamic graph learning, which belongs to a contemporary work with ours. The difference lies in that they rely on complex designs for handling co-occurrence neighbors of different nodes and temporal interval encoding. In contrast, we explore the feasibility of a simple Transformer for dynamic graphs without the need for complex modifications.

## 3 PRELIMINARIES

In this section, we first illustrate the problem of dynamic graph modeling. Then we briefly introduce the main components of Transformer architecture.

### 3.1 Dynamic Graph Modeling

We define a dynamic graph as $\mathcal{G} = (\mathcal{V}, \mathcal{E}, \mathcal{T}, \mathcal{X})$ with a set of nodes $\mathcal{V}$, edges $\mathcal{E}$, a time domain $\mathcal{T}$ and an input feature matrix $\mathcal{X}$. It can be characterized by a sequence of interacted links $\mathcal{G} = \{(v_i, v_j, \tau)_n : n = 1, 2, \ldots, |\mathcal{E}|\}$. Here, each tuple $(v_i, v_j, \tau)$ denotes a distinct interaction between nodes $v_i$ and $v_j$ at time $\tau \in \mathcal{T}$, with $|\mathcal{E}|$ representing the number of interactions within the temporal graph. Given the dynamic graph $\mathcal{G}$, we learn a model with parameter $\theta$ to capture the temporal evolution of the graph. The learned temporal representations can be used for different tasks such as node classification, link prediction and graph classification.

### 3.2 Transformer Architecture

The standard Transformer architecture comprises two main components: the multi-head self-attention layers (MHA) and the position-wise feed-forward network (FFN). In the following part, we will briefly introduce these blocks.

We represent an input sequence as $\mathbf{H} = \langle \mathbf{h}_1, \ldots, \mathbf{h}_N \rangle \in \mathbb{R}^{N \times d}$, where $d$ is the dimension of node features and $\mathbf{h}_i$ is the hidden representation for token $i$. The MHA module projects $\boldsymbol{H}$ to $H$ subspaces denoted as:

$$\boldsymbol{Q} = \boldsymbol{H}\boldsymbol{W}_Q, \boldsymbol{K} = \boldsymbol{H}\boldsymbol{W}_K, \boldsymbol{V} = \boldsymbol{H}\boldsymbol{W}_V, \tag{1}$$

where $\boldsymbol{W}_Q \in \mathbb{R}^{d \times d_K}$, $\boldsymbol{W}_K \in \mathbb{R}^{d \times d_K}$, $\mathbf{W}_V \in \mathbb{R}^{d \times d_V}$ re the learnable parameter matrices, and their dimensions are set as $d_K = d_V = d/H$.

The self-attention operation is performed using a scaled dot-product on the corresponding $(Q_h, K_h, V_h)$ for each head:

$$MHA(\boldsymbol{H}) = Concat(head_1, \ldots, head_H)\boldsymbol{W}_O,$$

$$head_h = Softmax(\frac{\boldsymbol{Q}_h \boldsymbol{K}_h^T}{\sqrt{d_K}})\boldsymbol{V}_h, \tag{2}$$

where $\boldsymbol{W}_O \in \mathbb{R}^{d \times d}$ is learnable parameter matrix.

The output of the MHA module is then passed through a Feed-Forward Network (FFN) layer followed by residual connection [12] and layer normalization (LN) [2]. Finally, the output of the $l^{th}$ layer

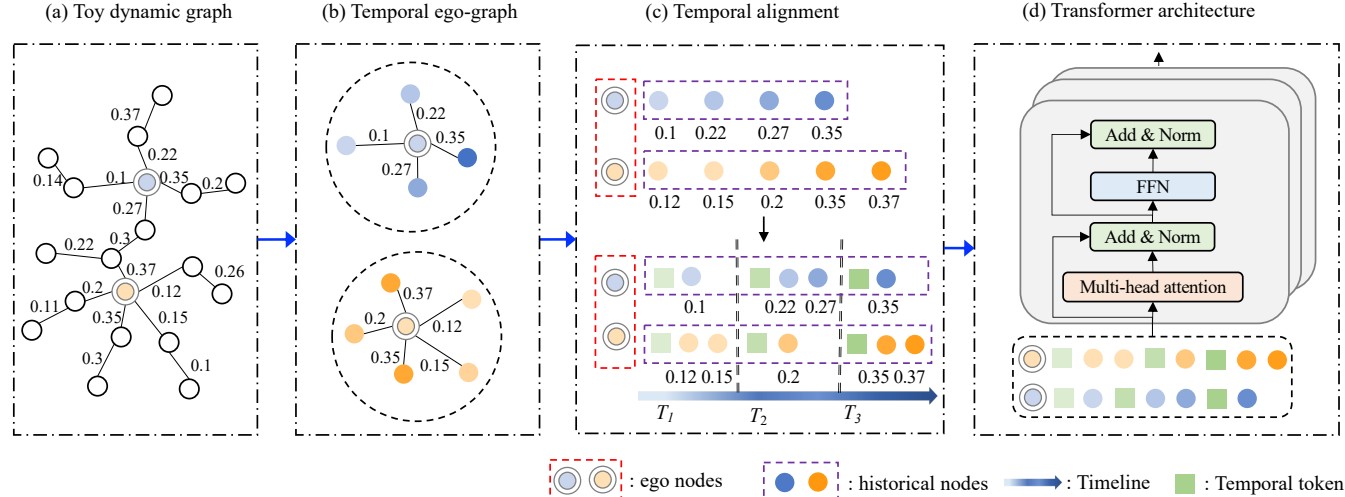

**Figure 2: Overall framework of SimpleDyG. (Best viewed in color. The numerical values adjacent to the links in (a) and (b), as well as beneath the nodes in (c), represent the time elapsed from the beginning, indicating the moments at which the links emerge (ranging from 0 to 1). The color intensity of nodes in the historical sequence represents the time span, where darker colors signify a longer-term duration, while lighter colors indicate a shorter-term duration.)**

$H^l$ is computed as follows:

$$\widehat{\boldsymbol{H}}^l = LN(\boldsymbol{H}^{l-1} + MHA(\boldsymbol{H}^{l-1})),$$
$$\boldsymbol{H}^l = LN(\widehat{\boldsymbol{H}}^l + FFN(\widehat{\boldsymbol{H}}^l)). \quad (3)$$

## 4 PROPOSED APPROACH

The overall framework of SimpleDyG is illustrated in Figure 3.1. Generally speaking, our framework is applied to a dynamic graph $\mathcal{G}$ (Figure 3.1(a)), where multiple temporal links emerge at various time points. In order to capture the dynamic evolution, we begin by extracting *temporal ego-graph* for ego-node which contains the entire historical interactions as shown in Figure 3.1(b). These temporal graphs are subsequently transformed into sequences while preserving their chronological order. To incorporate temporal alignment among different *ego-graphs*, we segment the timeline into various time spans with the same temporal interval as in Figure 3.1(c). Then we add *temporal tokens* into the ego-sequence to make our model identify different time spans. Finally, these sequences are fed into a Transformer architecture to facilitate various downstream tasks.

### 4.1 Temporal Ego-graph

As mentioned earlier, the sequence modeling capability of the Transformer architecture is well-suited for dynamic graph modeling. The strategy of mapping dynamic graphs into a sequence of tokens is crucial for the supported features and computational complexity. In this paper, we regard nodes in the dynamic graphs as input tokens which is a common approach in Transformer models for graphs. Besides, to preserve more historical interactions of all the nodes and ensure the scalability of dealing with large receptive fields, we extract the *temporal ego-graph* for each node in the dynamic graph. These temporal graphs are mapped into sequences to capture the structural and temporal evolution.

Specifically, we denote $v_i \in \mathcal{V}$ as an ego-node in the temporal graph $\mathcal{G}$. We extract the historically interacted nodes for $v_i$ and concatenate them into a sequence as input for Transformer architecture. Formally, we denote the *temporal ego-graph* for the ego-node $v_i$ as $w_i = \langle v_i^1, v_i^2 \ldots v_i^{|w_i|} \rangle$, where $|w_i|$ is the length of the historical interactions for node $v_i$. To better model the patterns within the input sequence, we follow similar practices as in NLP sequence modeling tasks and include some special tokens designed for our task. Finally, the input sequence and output sequence are constructed as follows [2]:

$$x_i = \langle |hist| \rangle, v_i, v_i^1, \ldots v_i^{|w_i|}, \langle |endofhist| \rangle,$$
$$y_i = \langle |pred| \rangle, v_i^{|w_i|+1}, \ldots, v_i^{|w_i|+z} \langle |endofpred| \rangle, \quad (4)$$

where the "$\langle |hist| \rangle$" and "$\langle |endofhist| \rangle$" are *special tokens* indicating the start and end of the input historical sequence. The "$\langle |pred| \rangle$" and "$\langle |endofpred| \rangle$" are reserved for predicting the next nodes at a future time. Specifically, the model will halt its predictions once the end *special token* is generated, enabling automatic decisions on the number of future interactions.

### 4.2 Temporal Alignment

In the original Transformer architecture, the input sequence is treated as a sequence of tokens, and the model captures the relationships between these tokens based on their relative positions in the sequence, representing temporal order information. However, it inherently lacks the capability to account for the universal time domain and the time interval and frequency information. In pursuit of this objective, we segment the time domain $\mathcal{T}$ into discrete, coarse-grained time steps, with each time step representing the same time interval, such as one week or one month, determined

---

[2]Special tokens in the beginning and at the end such as "$\langle |endoftext| \rangle$" are omitted for easy illustration.

by dataset characteristics. It's important to note that our approach differs from discrete-time graph modeling, as within each time step, we consider the precise temporal order of each link. We introduce a straightforward yet effective strategy to incorporate temporal alignment within dynamic graphs into the input sequence of the Transformer architecture. This strategy entails the use of special *temporal tokens* explicitly denoting different time steps that are globally recognized across all nodes. Suppose we split the time domain $\mathcal{T}$ into $T$ time steps, the sequence of ego-node $i$ in time step $t \in T$ is denoted as follows:

$$x'_i = \langle|hist|\rangle, v_i, \langle|time1|\rangle, S^1_i, \ldots \langle|timeT\text{-}1|\rangle, S^{T-1}_i, \langle|endofhist|\rangle,$$

$$y'_i = \langle|pred|\rangle, \langle|timeT|\rangle, S^T_i \langle|endofpred|\rangle, \tag{5}$$

$$S^t_i = \langle v^1_i, v^2_i \ldots v^{|S^t_i|}_i \rangle,$$

where $S^t_i$ represents the historical sequence of node $i$ as time step $t$ whith length of $|S^t_i|$. ($\langle|time1|\rangle \ldots \langle|timeT|\rangle$) are *temporal tokens* that serve as indicators of temporal alignment, allowing the model to recognize and capture temporal patterns in the data. By doing so, our approach enhances the Transformer's ability to understand the dynamics of the dynamic graph, making it more effective in tasks like predicting future interactions in social networks or other dynamic systems where temporal patterns are crucial.

## 4.3 Training objective

A training sample is formed by concatenating the input $x$ and output $y$ as $[x; y]$. We denote it as $r = \langle r_1, r_2, \cdots, r_{|r|}\rangle$ with $|r|$ tokens. For a given training instance in this format, we follow the original masking strategy, where, during the prediction of the $i$-th token, only the input sequence up to position $r_{<i}$ is taken into account, while the subsequent tokens are subject to masking. The joint probability of the next token is calculated as follows:

$$p(r) = \prod_{i=1}^{|r|} p(r_i|r_{<i}), \tag{6}$$

where $r_{<i}$ is the generated sequence before step $i$. $p(r_i|r_{<i})$ denotes the probability distribution of the token to be predicted at step $i$ conditioned with the tokens $r_{<i}$. It is computed as:

$$p(r_i|r_{<i}) = LN(\mathbf{R}^l_{<i})\boldsymbol{W}_{vocab}, \tag{7}$$

where $LN$ means layer normalization. $\mathbf{R}^L_{<i}$ denotes the hidden representation of the historically generated tokens before step $i$, which is obtained by the last layer of Transformer. $W_{vocab}$ is the learned parameter aiming to compute the probability distribution across the vocabulary of nodes in the graph.

Given a dataset containing $\mathcal{M}$ training instances, the loss function for training the model with parameters $\theta$ is defined as the negative log-likelihood over the entire training dataset as follows:

$$\mathcal{L} = -\sum_{m=1}^{|\mathcal{M}|} \sum_{i=1}^{n_m} log p_\theta(r^m_i|r^m_{<i}), \tag{8}$$

where $n_m$ is the length of the instance $r^m$.

We outline the training procedure of SimpleDyG in Algorithm 1. For each prediction step $i$ of one training instance, the hidden representations of the generated sequence $\mathbf{R}_{<i}$ are used for predicting

---

**Algorithm 1:** Training Procedure of SimpleDyG

**Input:** Dynamic graph $\mathcal{G} = (\mathcal{V}, \mathcal{E}, \mathcal{T}, \mathcal{X})$, training instances $\mathcal{M}$

**Output:** Well-trained model with parameter $\theta$ for dynamic graph modeling

**Initialization:** model parameter $\theta$

**while** *not converged* **do**
  sample a batch of instances $\mathcal{B}$ from $\mathcal{M}$
  **for** *each instance* $r = \langle r_1, r_2, \cdots, r_{|r|} \rangle$ *in batch* $\mathcal{B}$ **do**
    **while** *step* $i < |r|$ **do**
      /* prediction steps for one instance */
      Calculate the representation $\mathbf{R}_{<i}$ for $r_{<i}$
      Compute the joint probability by Equations 6 and 7
      Calculate the loss by Equation 8

**return** $\theta$, dynamic representation of $\mathcal{G}$

---

the next token. The joint probability of the next token is computed using Equations 6 and 7. Our model is trained using the Adam optimizer with a loss function based on negative log-likelihood, as presented in Equation 8.

## 5 EXPERIMENTS

### 5.1 Experimental Setup

**Datasets.** To evaluate the performance of our proposed method, we conducted experiments on four datasets from various domains, including the communication network UCI [30], the rating network ML-10M [11], the citation network Hepth [22], and the multi-turn conversation dataset MMConv [23]. The detailed statistics of all datasets after preprocessing are presented in Table 1.

**Table 1: Dataset statistics**

| Dataset | UCI | ML-10M | Hepth | MMConv |
|---------|-----|--------|-------|--------|
| Domain | Social | Rating | Citation | Conversation |
| # Nodes | 1,781 | 15,841 | 4,737 | 7,415 |
| # Edges | 16,743 | 48,561 | 14,831 | 91,986 |

**UCI** [30]: it represents a social network in which links represent messages exchanged among users. For temporal alignment, we and divide the dataset into 13 time steps following [36].

**ML-10M** [11]: we utilized the ML-10M dataset from MovieLens dataset comprising user-tag interactions, where the links connect users to the tags they have assigned to specific movies. For temporal alignment, the dataset is split into 13 time steps following [36].

**Hepth** [22]: it is a citation network related to high-energy physics theory. We extract 24 months of data from this dataset and split them into 12 time steps for temporal alignment. Note that this dataset contains new emerging nodes as time goes on. We use the extra word2vec [26] model to extract the raw feature for each paper based on the abstract.

**MMConv** [23]: this dataset contains a multi-turn task-oriented dialogue system that assists users in discovering places of interest across five domains. Leveraging this rich annotation, we represent

the dialogue as a dynamic graph which is also a widely studied approach in task-oriented dialogue systems. For temporal alignment, we empirically divided the dataset into 16 time steps, each corresponding to a distinct turn in the conversation.

**Baselines.** We compare our method with baselines in two categories: (1) discrete-time approaches: DySAT [36] and EvolveGCN [31] (2) continuous-time approaches: DyRep [40], JODIE [20], TGAT [48], TGN [35] and GraphMixer [6].

- **DySAT** [36] leverages joint structural and temporal self-attention to learn the node representations at each timestep.
- **EvolveGCN** [31] adapts to evolving graph structures by employing RNN to evolve graph convolutional network parameters.
- **DyRep** [40] utilizes a two-time scale deep temporal point process model to capture temporal graph topology and node activities.
- **JODIE** [20] focuses on modeling the binary interaction among users/items by two coupled RNNs. A projection operator is designed to predict the future representation of a node at any time.
- **TGAT** [48] employs temporal graph attention layers and time encoding techniques to aggregate temporal-topological features.
- **TGN** [35] combines the memory modules and message-passing to maintain the dynamic representations. This model also adopts time encoding and temporal graph attention layers.
- **GraphMixer** [6] relies on MLP layers and neighbor mean-pooling to learn the link and node encoders. An offline time encoding function is adopted to capture the temporal information.

**Implementation Details.** In this paper, we evaluate the performance of SimpleDyG on the link prediction task. Given the ego-nodes, the objective of the link prediction task is to predict the possible linked nodes at time step $T$. For all the datasets, we follow the setting in [6] by treating the temporal graphs as undirected graphs. We split each dataset into training/validation/testing based on the predefined time steps. We choose the data at the last time step $T$ as the testing set, while the data at time step $T - 1$ serves as the validation set, with the remaining data for training. We tune the parameters for all methods on the validation set. All experiments are repeated ten times, and we report the averaged results with standard deviation. We provide further implementation details and hyper-parameter settings for the baselines in Appendix A and B.

**Evaluation Metrics.** In our evaluation, we carefully selected metrics that are well-suited to our specific task. The goal of the link prediction task is to predict a set of nodes linked to each ego-node. Notably, our SimpleDyG model predicts a node sequence, with each prediction influenced by the prior ones until the generation of an end token. In contrast, the baseline models make simultaneous predictions of entire ranking sequences for each ego-node. To evaluate ranking performance and set similarity between predicted and ground truth node sets, we employ two key metrics: *NDCG@5* and *Jaccard* similarity. *NDCG@5* is a well-established metric commonly used in information retrieval and ranking tasks [43], aligning with our objective of ranking nodes and predicting the top nodes linked to an ego-node. On the other hand, *Jaccard* similarity is valuable for quantifying the degree of overlap between two sets [13], measuring the similarity between predicted nodes and the ground truth nodes associated with the ego-node. Specifically, for the baseline models,

we choose the top $k$ nodes ($k = 1, 5, 10, 20$) as the predicted set, as they are not generation models and cannot determine the end of the prediction. We then select the maximum Jaccard similarity value across different $k$ as the final Jaccard similarity score. This comprehensive evaluation strategy ensures a thorough assessment of our models and baselines in predicting linked nodes.

## 5.2 Main Performance Comparison

We report the results of all methods under *NDCG@5* and *Jaccard* across four diverse datasets in Table 2. Generally speaking, our method outperforms all the baselines on all datasets, and we make the following key observations.

Firstly, we find that continuous-time approaches generally perform better than discrete ones across a wide range of scenarios, indicating the important role of time-related information in dynamic graph analysis. Notably, continuous-time baselines such as GraphMixer exhibit superior performance. This superiority can be mainly attributed to the simple MLP-Mixer architecture, which makes it easier to capture long-term historical sequences with lower complexity. In contrast, other models like DyRep, TGAT, and TGN, which rely on complex designs such as GNNs and GATs, display subpar performance. This phenomenon stems from the inherent limitations of GNNs and GATs in capturing distant relationships or broader historical contexts within predefined time windows.

Secondly, for the inductive scenarios such as the Hepth dataset, the models deployed by GNNs, GATs, and Transformer show advanced performance. Their effectiveness lies in their ability to capture intricate patterns and relationships within dynamic graphs, especially when faced with inductive scenarios. In contrast, simpler models may struggle to adapt to these situations, resulting in suboptimal performance.

In summary, while continuous-time approaches have generally been shown effective, it is essential to consider the specific characteristics of the application scenario and strike a delicate balance between model complexity and the necessity to capture long-range dependencies in dynamic graph modeling.

## 5.3 Effect of Extra Tokens

We design extra tokens to make the vanilla Transformer architecture more suitable for dynamic graph modeling. To assess their effectiveness, we conduct an in-depth analysis of these token designs including *special tokens* indicating the input and output, and the *temporal tokens* for aligning among *temporal ego-graphs*.

**Impact of the *special tokens*.** The *special tokens* include the start and end of the historical sequence ("$\langle|hist|\rangle$" and "$\langle|endofhist|\rangle$"), as well as the predicted sequence ("$\langle|pred|\rangle$" and "$\langle|endofpred|\rangle$"). To comprehensively evaluate their effect across diverse scenarios, we examine two degenerate variants: (1) *same special*, where we use the same *special tokens* for input and output. (2) *no special*, where we entirely removed all *special tokens* from each sample. We show the results in Table 3 and make the following observations.

In general, *special tokens* enhance the link prediction performance across different datasets. Furthermore, the differences between the *same special* and original *SimpleDyG* tend to be minimal. However, an interesting finding emerges in the case of the Hepth dataset, where the *no special* scenario yields the best performance.

**Table 2: Performance of dynamic link prediction by SimpleDyG and the baselines on four datasets.(In each column, the best result is bolded and the runner-up is underlined. "-" indicates the method is not suitable for inductive scenario.)**

| | UCI | | ML-10M | | Hepth | | MMConv | |
|---|---|---|---|---|---|---|---|---|
| | *NDCG@5* | *Jaccard* | *NDCG@5* | *Jaccard* | *NDCG@5* | *Jaccard* | *NDCG@5* | *Jaccard* |
| DySAT [36] | 0.010±0.003 | 0.010±0.001 | 0.058±0.073 | 0.050±0.068 | - | - | 0.102±0.085 | 0.095±0.080 |
| EvolveGCN [31] | 0.064±0.045 | 0.032±0.026 | 0.097±0.071 | 0.092±0.067 | 0.009±0.004 | 0.007±0.002 | 0.051±0.021 | 0.032±0.017 |
| DyRep [40] | 0.011±0.018 | 0.010±0.005 | 0.064±0.036 | 0.038±0.001 | 0.031±0.024 | 0.010±0.006 | 0.140±0.057 | 0.067±0.025 |
| JODIE [20] | 0.022±0.023 | 0.012±0.009 | 0.059±0.016 | 0.020±0.004 | 0.031±0.021 | 0.011±0.008 | 0.041±0.016 | 0.032±0.022 |
| TGAT [48] | 0.061±0.007 | 0.020±0.002 | 0.066±0.035 | 0.021±0.007 | 0.034±0.023 | 0.011±0.006 | 0.089±0.033 | 0.058±0.021 |
| TGN [35] | 0.041±0.017 | 0.011±0.003 | 0.071±0.029 | 0.023±0.001 | 0.030±0.012 | 0.008±0.001 | 0.096 ±0.068 | 0.066±0.038 |
| GraphMixer [6] | **0.104±0.013** | 0.042±0.005 | 0.081±0.033 | 0.043±0.022 | 0.011±0.008 | 0.010±0.003 | 0.172±0.029 | 0.085±0.016 |
| SimpleDyG | **0.104±0.010** | **0.092±0.014** | **0.138±0.009** | **0.131±0.008** | **0.035±0.014** | **0.013±0.006** | **0.184±0.012** | **0.169±0.010** |

**Table 3: Impact of *special tokens* in SimpleDyG across four datasets.**

| | UCI | | ML-10M | | Hepth | | MMConv | |
|---|---|---|---|---|---|---|---|---|
| | *NDCG@5* | *Jaccard* | *NDCG@5* | *Jaccard* | *NDCG@5* | *Jaccard* | *NDCG@5* | *Jaccard* |
| *SimpleDyG* | 0.104±0.010 | 0.092±0.014 | **0.138±0.009** | **0.131±0.008** | 0.035±0.014 | 0.013±0.006 | **0.184±0.012** | 0.169±0.010 |
| *same special* | **0.113±0.007** | **0.095±0.010** | 0.085±0.046 | 0.079±0.043 | 0.027±0.014 | 0.009±0.005 | 0.179±0.013 | **0.170±0.010** |
| *no special* | 0.041±0.025 | 0.020±0.011 | 0.006±0.009 | 0.006±0.009 | **0.096±0.016** | **0.025±0.006** | 0.01±0.008 | 0.008±0.007 |

**Table 4: Impact of different temporal alignment designs on the four datasets.**

| | UCI | | ML-10M | | Hepth | | MMConv | |
|---|---|---|---|---|---|---|---|---|
| | *NDCG@5* | *Jaccard* | *NDCG@5* | *Jaccard* | *NDCG@5* | *Jaccard* | *NDCG@5* | *Jaccard* |
| *SimpleDyG* | 0.104±0.010 | **0.092±0.014** | 0.138±0.009 | 0.131±0.008 | 0.035±0.014 | 0.013±0.006 | 0.184±0.012 | 0.169±0.010 |
| *same time* | 0.09±0.013 | 0.083±0.012 | **0.147±0.001** | **0.139±0.001** | **0.046±0.009** | 0.017±0.004 | 0.24±0.031 | 0.212±0.025 |
| *no time* | **0.111±0.015** | 0.091±0.014 | 0.117±0.062 | 0.111±0.059 | 0.045±0.007 | **0.018±0.003** | **0.26±0.019** | **0.237±0.016** |

It can be explained by the specific character of the citation dataset. In the testing data of Hepth, the ego-nodes are all newly emerged nodes indicating the newly published papers. Consequently, the input samples lack any historical information, leaving the distinction between history and the future meaningless.

**Impact of *temporal tokens*.** To comprehensively evaluate the impact of *temporal tokens*, we compare the performance with two degenerate variants: (1) *same time*, where we do not distinguish specific time steps and employ the same *temporal tokens* for each time step. (2) *no time*, in which we entirely removed all *temporal tokens* from each sample. The results are presented in Table 4 and we have the following observations.

It is surprising and interesting to observe performance improvement with a simpler design for temporal alignment. This phenomenon is most noticeable in the MMConv and Hepth datasets due to the characteristics of these datasets. The citation relationship and the conversation among different ego-nodes do not strictly follow temporal segmentation. Using the same *temporal tokens* or none at all allows the model to adapt more naturally to this temporal order.The temporal alignment plays an important role for UCI and

ML-10M datasets. However, they show different trends with the *same time* version. The reason is that the communication habits of different users are more related to temporal information, while the rating habits of users are more subjective in ML-10M dataset.

## 5.4 The Performance of Multi-step Prediction

We evaluate the ability of SimpleDyG for multi-step prediction with the time steps range from $t$ to $t + \triangle t$, utilizing a model that has been trained on data up to time $t$. Here, the time step means the coarse segment of the time domain as did in the temporal alignment. In our experiment, we set $\triangle t$ as three and achieve multi-step prediction step by step constrained by the results of previous steps. The performance trends of SimpleDyG with two baselines TGAT and GraphMixer are illustrated in Figure 3.

We observe a natural decay in performance over time for all methods, as anticipated. However, what stands out is SimpleDyG's ability to consistently outperform the baselines as time progresses. This observed trend underscores the effectiveness of our proposed Transformer architecture in modeling dynamic graph data. Notably, different datasets exhibit varying patterns of performance decay

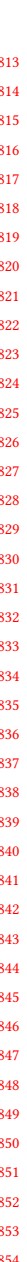

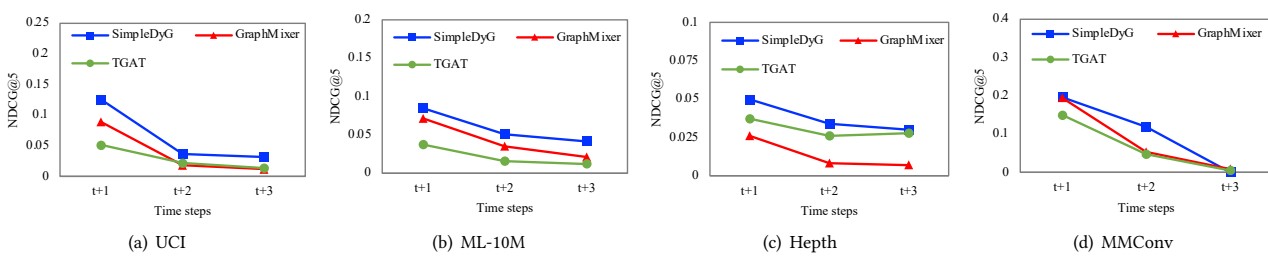

(a) UCI          (b) ML-10M          (c) Hepth          (d) MMConv

**Figure 3: The performance of multi-step prediction.**

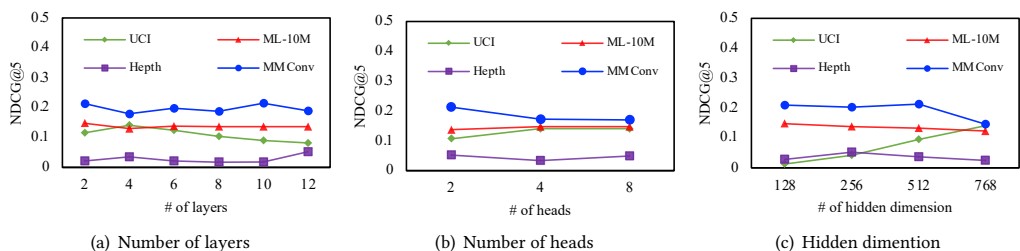

(a) Number of layers          (b) Number of heads          (c) Hidden dimention

**Figure 4: Impact of hyper-parameters.**

over time, highlighting the importance of dataset-specific considerations in dynamic graph analysis. For instance, in the case of the ML-10M and Hepth datasets, we notice a relatively slight performance drop over time. This phenomenon can be attributed to specific characteristics inherent to these datasets. In the ML-10M dataset, the presence of numerous historical interactions contributes to a relatively stable performance trend. The dataset's richness in historical data allows the model to absorb small noise or fluctuations without a significant impact on overall performance. On the other hand, the Hepth dataset introduces a unique challenge due to the presence of new nodes at each time step. Despite this inherent complexity, SimpleDyG still demonstrates its adaptability by maintaining competitive performance, reflecting its capability to adapt to dynamic scenarios and evolving graph structures.

## 5.5 Hyper-parameter Analysis

We undertake an examination of the critical hyper-parameter choices, taking into account the variations observed across different datasets. Specifically, we systematically explore the impact of several crucial hyper-parameters, namely the number of layers, the number of heads, and the hidden dimension size. These hyper-parameters play a pivotal role in shaping the model's capacity and its ability to capture intricate patterns within dynamic graphs. We fine-tune the hyper-parameters while keeping all other parameters constant. From Figure 4, we draw some highlights as follows:

- **Number of layers**: The variance of performance under different numbers of layers is relatively small. This suggests that the choice of the number of layers in SimpleDyG has a more consistent impact across different datasets and scenarios. Generally speaking, two layers are typically sufficient for most cases. For inductive scenarios such as the Hepth dataset, it is advisable to use more layers to effectively capture the evolving graph structure.

- **Number of heads**: For the number of attention heads, we find that using either 2 or 4 heads is generally suitable for a wide range of scenarios. These settings provide a good balance between performance and computational efficiency.

- **Hidden dimension size**: The choice of hidden dimension size depends on the complexity of the dataset. For datasets like movie ratings (e.g., ML-10M), a hidden dimension size of 128 is often adequate. However, for datasets involving more intricate interactions, such as communication networks or conversation datasets, it becomes necessary to use larger hidden dimension sizes like 256 or 512. Notably, the UCI dataset requires a hidden dimension of 768, which can be explained by the complexity and richness of the interactions among users within the dataset.

## 6 CONCLUSION

In this work, we've delved into the intricate realm of dynamic graph modeling, recognizing its profound significance across a range of applications. Drawing from the strengths of the Transformer's self-attention mechanism, we tailored a solution that sidesteps the often convoluted designs prevalent in existing methods. Our novel approach re-envisions dynamic graphs from a sequence modeling perspective, leading to the development of an innovative temporal alignment technique. This strategic design not only adeptly captures the temporal dynamics inherent in evolving graphs but also simplifies their modeling process. Our empirical investigations, carried out across four real-world datasets spanning diverse sectors, serve as a testament to our model's efficacy. In the future, we will delve deeper into the nuances of the temporal alignment technique for further optimizations. Additionally, the potential for integrating more advanced attention mechanisms can be explored to further elevate the model's capabilities in capturing dynamic evolutions.

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

# A  ADDITIONAL IMPLEMENT DETAILS

Note that the implementation details of baseline approaches in their publicly released code are quite different. For instance, most of them regard the link prediction task as binary classification, where the objective is to determine the presence or absence of links between the positive pairs of nodes and randomly selected negative pairs. They either employ a binary cross-entropy loss to facilitate classifier learning or utilize logistic regression to train an additional classifier. To tailor these baselines to our specific task for a fair comparison, we adapt them into a ranking task and substitute the classifier loss with a pair-wise Bayesian personalized ranking (BPR) loss for all baselines.

# B  HYPER-PARAMETERS SETTINGS OF BASELINES

Considering that we refine the loss of the baselines as BPR loss, we tune the important parameters of all baselines for all the datasets. For all baselines, we tune the parameter of hidden dimension with $\{16, 32, 64, 128, 256, 512\}$ for each dataset. For a fair comparison with our model, we don't set a historical window for discrete-time approaches and use all the historical data.

Some important parameters for each baseline are listed as follows: For DySAT [36], We set the self-attention layers and head to be 2 and 8, respectively. For EvolveGCN [31], the number of GCN layers is 1. For DyRep [40], the message aggregation layer is 2, and the number of neighbor nodes is 20. For TGAT [48] and TGN [35], the number of graph attention heads is 2 and the attention layers are 1 and 2, respectively. For GraphMixer [6], the number of MLP layers for UCI is 1 and 2 for other datasets. For a fair comparison, we set the historical length of each node to 1024, which is the same as our model.

