# OpenReview forum: "On the Feasibility of Simple Transformer for Dynamic Graph Modeling"
_ACM.org/TheWebConf/2024/Conference — TheWebConf24_

### Official Review · Reviewer_6j6U · 2023-11-23

**Novelty:** 5
**Technical Quality:** 4

**Review:**

It introduces a novel strategy for tokenizing dynamic graphs for Transformer-based architectures. Based on the strategy, the proposed method outperforms the existing representation learning methods for dynamic graphs while it does not require complicated architecture or a heavy amount of computation. In addition, The paper is well-written and easy to read. However, the proposed method only supports incremental settings and cannot handle fully dynamic graphs with the deletion of existing links.

**Questions:**

- There is a typo: Figure 3.1 -> Figure 2?
- It would be better if the proposed model supports link deletion operation. It seems that the current version only supports link insertions.
- The authors need to provide the reason why DySAT is not suitable for inductive scenarios on the Hepth dataset.
- The considered datasets are too small and all the reported metric scores are too low (<0.25). Could you provide the reasons?
- As shown in Table 4, using the current design related to temporal tokens is not so effective, compared to the other designs provided in the table.

**Reviewer Confidence:**

3: The reviewer is confident but not certain that the evaluation is correct

**Scope:**

3: The work is somewhat relevant to the Web and to the track, and is of narrow interest to a sub-community

---

### Official Review · Reviewer_CFgg · 2023-11-24

**Novelty:** 4
**Technical Quality:** 4

**Review:**

This paper re-conceptualizes dynamic graphs as a sequence modeling challenge and introduces an innovative temporal alignment technique. This technique not only captures the inherent temporal evolution patterns within dynamic graphs but also streamlines the modeling process of their evolution.
Strengths:
Propose a simple yet surprisingly effective Transformer-based approach for dynamic graphs, called SimpleDyG, without complex modifications.
Introduce a novel strategy to map a dynamic graph into a set of sequences, by considering the history of each node as a temporal ego-graph.
Weaknesses:
--Lack of the time complexity analysis and the corresponding time efficiency experiments.
--Lack of the baselines proposed in recent years.
-- Explanation is needed for the superscript in Equation (8).
--It is better to add some ablation studies.
--Some typos, such as ‘we and divide’

**Questions:**

Some dataset split into different time steps following[36]. Is there real dynamic network dataset?

**Reviewer Confidence:**

3: The reviewer is confident but not certain that the evaluation is correct

**Scope:**

3: The work is somewhat relevant to the Web and to the track, and is of narrow interest to a sub-community

---

### Official Review · Reviewer_WECH · 2023-11-24

**Novelty:** 2
**Technical Quality:** 4

**Review:**

The paper focuses on utilizing the Transformer model for dynamic graph modeling. The authors challenge the existing complex methodologies in dynamic graph modeling, which often neglect detailed temporal aspects and struggle with long-term dependencies. Their proposed method, SimpleDyG, leverages the self-attention mechanism of Transformers to handle these long-range dependencies without intricate modifications. This approach reconceptualizes dynamic graphs as sequence modeling challenges and introduces a temporal alignment technique to capture temporal evolution patterns. The effectiveness of this model is demonstrated through experiments on various real-world datasets.

Pros：

The paper proposes a novel method of applying the Transformer architecture, primarily used in NLP and CV, to dynamic graph modeling.

SimpleDyG's utilization of the inherent self-attention mechanism in Transformers without complex modifications stands out as a strength.

Cons:

1. The experimental evaluation is not sufficient. a) Better to adopt more datasets, like datasets in [1-2]. b) Missing important dynamic GNN baselines.

2. The limited novelty is a major concern. The authors claim that 'all these previous Transformer-based approaches only focus on static graphs, leaving unanswered questions about the feasibility for dynamic graphs', but there already exist several works about dynamic graph transformers[2-6]. It is a misclaim, and the authors are expected to tell the difference between this paper and these papers.

[1] Huang, Shenyang, et al. "Temporal graph benchmark for machine learning on temporal graphs." arXiv preprint arXiv:2307.01026 (2023).

[2] Yu, Le, et al. "Towards Better Dynamic Graph Learning: New Architecture and Unified Library." arXiv preprint arXiv:2303.13047 (2023).

[3] Wang, Lu, et al. "Tcl: Transformer-based dynamic graph modelling via contrastive learning." arXiv preprint arXiv:2105.07944 (2021).

[4] Liu, Yixin, et al. "Anomaly detection in dynamic graphs via transformer." IEEE Transactions on Knowledge and Data Engineering (2021).

[5] Cong, Weilin, et al. "Dynamic graph representation learning via graph transformer networks." (2021).

[6] Wang, Zehong, et al. "Temporal graph transformer for dynamic network." International Conference on Artificial Neural Networks. Cham: Springer Nature Switzerland, 2022.

**Questions:**

see weaknesses

**Reviewer Confidence:**

4: The reviewer is certain that the evaluation is correct and very familiar with the relevant literature

**Scope:**

4: The work is relevant to the Web and to the track, and is of broad interest to the community

---

### Official Review · Reviewer_bUWR · 2023-12-01

**Novelty:** 5
**Technical Quality:** 5

**Review:**

**Summary**

The paper introduces a straightforward yet effective transformer-based model tailored for dynamic graph modeling. This method excels in capturing the inherent temporal evolution patterns across an entire timeline, effectively addressing the challenges posed by dynamic graphs. The key innovation lies in transforming a dynamic graph into a sequence of data points, which are then processed using a transformer model. The authors support their methodology with extensive experiments across various datasets, demonstrating the proposed model's superior performance in capturing the evolving patterns in dynamic graphs.

**Pros**

1. The paper is notably well-written, offering clear and detailed explanations complemented by illustrative figures that effectively outline the algorithm's pipeline. These visuals significantly enhance comprehension, making the complex methodologies and concepts more accessible to readers.
2. The approach presented in the paper is both simple and highly effective. The authors introduce an intelligent method to incorporate graph information into a sequence of tokens, making it particularly suitable for dynamic graphs. This technique, specifically using transformers, excels in capturing granular temporal information.

**Cons**

In the experimental section, providing additional statistics about the temporal patterns in each dataset would greatly enhance the paper. For example, whether the changes in the datasets are gradual or dramatic would be particularly useful. Insight into the specific temporal characteristics of each dataset can offer a clearer view of how the proposed method performs under varying temporal dynamics.

**Questions:**

In Section 5.2, it's noted that the Hepth dataset is considered a more inductive scenario. Could the authors elaborate on the rationale behind this characterization? Providing a deeper understanding of why the Hepth dataset fits this scenario would aid in comprehending how different types of algorithms perform across various datasets.

**Reviewer Confidence:**

3: The reviewer is confident but not certain that the evaluation is correct

**Scope:**

4: The work is relevant to the Web and to the track, and is of broad interest to the community

---

### Decision · Program_Chairs · 2024-01-22

**Decision:**

Accept

**Comment:**

There was some discussion on this paper and I thought quite a bit about it.

 The paper proposes a simple architecture and demonstrate that transformers work for temporal graph modeling. I think the simplicity is a plus, especially if previous work was more complicated (which is what I gather from the reviews). The experiments are on fairly small datasets, so I'm not sure how scalable the method is. But the experiments seem quite comprehensive,

 I'll rate as a weak accept paper, though I could also see it as borderline.